Cardiotoxicity detection tool for breast cancer chemotherapy: a retrospective study

Alenezi Ahmad 1 ahmad.alenezi1@ku.edu.kw
McKiddie Fergus 2
Nath Mintu 3
Mayya Ali 4
http://orcid.org/0000-0002-6350-0673 Welch Andy 5
1 Department of Radiologic Sciences, Kuwait University , Jabriya , Kuwait
2 Nuclear Medicine, Unaffiliated , Aberdeen, Aberdeenshire , United Kingdom
3 Institute of Applied Health Sciences, University of Aberdeen , Aberdeen , United Kingdom
4 Department of Computers and Automatic Control Engineering, Tishreen University , Latakia , Syria
5 Medical and Dental Sciences Department, University of Aberdeen , Aberdeen , United Kingdom
Alatas Bilal
Electronic publication date: 2024 Aug 2
Publication date: 2024
Volume: 12
Electronic Location ID: e2230
Received 2024 Jan 11; Accepted 2024 Apr 25
Copyright: © 2024 Alenezi et al.
Copyright year: 2024
Copyright holder: Alenezi et al.
License: This is an open access article distributed under the terms of the Creative Commons Attribution License, which permits unrestricted use, distribution, reproduction and adaptation in any medium and for any purpose provided that it is properly attributed. For attribution, the original author(s), title, publication source (PeerJ) and either DOI or URL of the article must be cited.
License URL: https://creativecommons.org/licenses/by/4.0/

Keywords: LVEF, MATLAB, Dyssynchron, Cancer, Processing

Funding: The authors received no funding for this work.

==============================
Background

Patients with breast cancer undergoing biological therapy and/or chemotherapy perform multiple radionuclide angiography (RNA) or multigated acquisition (MUGA) scans to assess cardiotoxicity. The association between RNA imaging parameters and left ventricular (LV) ejection fraction (LVEF) remains unclear.

Objectives

This study aimed to extract and evaluate the association of several novel imaging biomarkers to detect changes in LVEF in patients with breast cancer undergoing chemotherapy.

Methods

We developed and optimized a novel set of MATLAB routines called the “RNA Toolbox” to extract parameters from RNA images. The code was optimized using various statistical tests (e.g., ANOVA, Bland-Altman, and intraclass correlation tests). We quantitatively analyzed the images to determine the association between these parameters using regression models and receiver operating characteristic (ROC) curves.

Results

The code was reproducible and showed good agreement with validated clinical software for the parameters extracted from both packages. The regression model and ROC results were statistically significant in predicting LVEF (R2 = 0.40, P < 0.001) (AUC = 0.78). Some time-based, shape-based, and count-based parameters were significantly associated with post-chemotherapy LVEF (β = 0.09, P < 0.001), LVEF of phase image (β = 4, P = 0.030), approximate entropy (ApEn) (β = 11.6, P = 0.001), ApEn (diastolic and systolic) (β = 39, P = 0.002) and LV systole size (β = 0.03, P = 0.010).

Conclusions

Despite the limited sample size, we observed evidence of associations between several parameters and LVEF. We believe that these parameters will be more beneficial than the current methods for patients undergoing cardiotoxic chemotherapy. Moreover, this approach can aid physicians in evaluating subclinical cardiac changes during chemotherapy, and in understanding the potential benefits of cardioprotective drugs.

Introduction

The process of understanding and predicting the cardiotoxicity of chemotherapy has been extensively researched (Rüger et al., 2020). Assessing the impact of cardiotoxicity on the function of the left ventricle (LV) is crucial, as it serves as a significant biomarker for the integrity of the LV (Rüger et al., 2020). Ventricular function can be assessed using various imaging modalities, including magnetic resonance imaging, echocardiography (cardiac ultrasound, including Doppler techniques) (Benmalek et al., 2021), and nuclear medicine imaging, such as single-photon emission tomography (SPECT) and positron emission tomography (PET) (Boyd et al., 1996; Di Carli et al., 1994; Fukunaga et al., 2021; Jones et al., 2020; Régis & Rouzet, 2022; Singh & Jindal, 2020; Singh et al., 2013; Suzuki et al., 2016). The LV can also be assessed using radionuclide angiography (RNA) scans to evaluate various parameters (Mitra & Basu, 2012), which can be derived retrospectively without additional radiation exposure. Although some of these parameters have been investigated, they have not yet been implemented in routine practice owing to their novelty or the existence of limited published data in the context of nuclear medicine imaging (Jones et al., 2020). In fact, left ventricle ejection fraction (LVEF) is a widely recognized marker for early detection of cardiac dysfunction, holds paramount importance in the assessment of cardiotoxicity, serving as a key indicator for monitoring changes in left ventricular performance and guiding clinical interventions according to established literature in the field (Suzuki et al., 2016).

Jones et al. (2020) analyzed approximate entropy (ApEn) to evaluate ventricular function in patients with breast cancer undergoing cardiotoxic chemotherapy, and patients with a higher ApEn before treatment showed a decreased LVEF following chemotherapy. Thus, ApEn may be useful for detecting chemotherapy-related cardiac dysfunction (CTRCD) because it may predict LVEF decline.

Dyssynchrony is the most widely investigated parameter of the LV function. Synchrony has been used to diagnose and evaluate treatment outcomes of cardiac resynchronization therapy (CRT). Several studies have demonstrated that synchrony is a reproducible and objective tool for mechanical diagnosis. Table 1 summarizes various studies investigating dyssynchrony parameters (Badhwar et al., 2016; Beneyto et al., 2021; Jiménez-Ángeles et al., 2006; Johnson et al., 2015; Lalonde et al., 2010, 2014; Nakajima et al., 2017; Nichols et al., 2010; O’Connell et al., 2005). However, to date, no studies have implemented all dyssynchrony and shape parameters (e.g., elongation and circularity) within a single predictive model.

Table 1 Dyssynchrony studies in literature.

Study	Modality	Parameter	Results	Disadvantages	
Johnson et al. (2015)	SPECTa, Echo, MRI	Eb & Sc	Echo, MRI and RNA right ventricular parameters can distinguish patients with ARCVd from controls with structurally normal hearts.	Small sample number and LVe segmentation method seemed to be manual	
Beneyto et al. (2021)	Planner/SPECT RNAf	PSDg	Abnormal PSD was strongly associated with the occurrence of ventricular arrythmia.	Used fully automated image processing	
Lalonde et al. (2014)	SPECT	PSD, E & S	Mechanical dyssynchrony was detected using cluster analysis and produced equivalent results to that from FFTh, E and S after optimizing phase parameters.	Potential bias due to relatively small sample number. Gating was limited to eight, which may decrease accuracy.	
Jiménez-Ángeles et al. (2006)	RNA	Factorial phase image, S	There are significant differences between Fourier phase image and factorial phase image that may affect S & E readings. Mean LV S and E were 0.99 and 0.79, respectively.	Focused on normal subject. No specificity and sensitivity data are available	
O’Connell et al. (2005)	RNA	Pi, S and E	Using simulation data, S and E were proven to be able to objectively measure regional ventricular synchrony.	Very limited clinical data.	
Badhwar et al. (2016)	RNA	IVSj, LVSk	IVS and LVS can predict positive response of cardiac resynchronization therapy.	Only included in-vitro radiolabeling technique for mostly males.	
Nakajima et al. (2017)	MPIl	PAm, S	Compared PSD between four different software. There was dependency on gender and LV volume and LVEFn when comparing PSD (i.e., phase parameter showed larger variation in patient with low LVEF and high LV volume)	Conducted retrospectively on normal patients using multi-center and multiple software without clear protocol.	
Nichols et al. (2010)	RNA (SPECT)	E, S	E and S was able to detect apical dyskinesia	Used manual segmentation (reproducibility concerns).	
Lalonde et al. (2010)	RNA (SPECT)	PSD, S, E	PSD, E and S were able to diagnose mechanical dyssynchrony in a reproducible fashion	Fixed ROIso and Edge detection algorithm threshold value was not monitored.	
Note:

a SPECT, single-photon emission tomography; b E, entropy; c S, synchrony; d ARCV, arrhythmogenic right ventricular cardiomyopathy; e LV, left ventricle; f RNA, radionuclide angiography; g PSD, phase standard deviation; h FFT, first Fourier transform; I P, phase; j IVS, intraventricular synchrony; k LVS, left ventricle synchrony; m PA, phase angle; n LVEF, left ventricle ejection fraction; o ROIs, regions of interest.

In this study, we developed a novel package of codes, the “RNA Toolbox” (Alenezi, 2022) to derive potentially useful parameters of cardiotoxicity using semi-automated techniques to enhance reliability. MATLAB was employed to develop codes performing core functions such as filtering, correction, and segmentation (using phase and amplitude images), to extract features from the RNA images. This research attempted to build on the previous understanding of some of these parameters (e.g., LVEF), as well as the novel parameters of entropy (E), approximate entropy (ApEn), bounded-E, synchrony, lung-to-heart ratio (LHR), phase LVEF (i.e., LVEF derived from phase images), circularity, elongation, short/long-axis fractional shortening (FS), and beat duration. The RNA Toolbox was developed to process RNA images and derive all parameters using a single processing step. The definitions of the corresponding parameters are presented in Table S1.

To accurately derive these parameters, optimization of the RNA Toolbox is imperative (Fig. 1). The optimization process consisted of image correction, followed by segmentation. Segmentation of images is problematic, owing to the inherently excessive noise in RNA images. Therefore, we hypothesized a novel segmentation method, ACRG (active-contouring region-growing), that combined both region-growing (RG) and active-contouring (AC) algorithms and was applied to phase and amplitude images (see the graphical abstract Fig. 1). Notably, the intention of this study was not to compare multiple modalities, but rather to compare algorithms within the same modality (i.e., Gamma Camera).

Figure 1 Steps of the research plan.

Step (1): Write and develop the code to analyze the images. Step (2): Code optimization process. This is composed of background optimization to correct images (2a), a filtering step to enhance images (2b), and image segmentation to define the left ventricle using a novel algorithm (2c). Step (3): Code implantation. From this, parameters were derived and predictive parameters were used in the regression model.

Materials and Methods

Code development

In this study, we developed a novel package of codes, the “RNA Toolbox” to derive potentially useful parameters of cardiotoxicity using semi-automated techniques to enhance reliability. MATLAB was employed to develop codes performing core functions such as filtering, correction, and segmentation (using phase and amplitude images), to extract features from the RNA images.

Dataset

We retrospectively analyzed 110 patients with breast cancer who were treated with various cardiotoxic agents. Patients younger than 75 years with good RNA image quality, no missing clinical information, and who were not terminally ill from breast cancer were included in the study. Patients with cardiac illnesses (n = 1), with breast cancer beyond stage 3 (n = 3) or underwent radiotherapy (n = 3) were excluded from the study. Seventy-seven patients underwent cardiotoxic 5 Fluorouracil (5FU), Epirubicin, and Cyclophosphamide (FEC) chemotherapy, while the remaining patients received a combination of trastuzumab biological therapy and FEC therapy at Aberdeen Royal Infirmary, NHS Grampian, Aberdeen, UK. All patients underwent a baseline scan, and one or more RNA scans were performed after chemotherapy. All RNA images were assessed using a validated clinical package called eSoft® (Siemens Healthcare GmbH, Erlangen, Germany), along with the in-house MATLAB® RNA Toolbox (The Math Works, Inc. Natick, NY, USA; Version 2020a). This study was approved by the Institutional Ethics Review Board of the University of Aberdeen and Aberdeen Royal Infirmary, NHS Grampian. The requirement for informed consent was waived due to the retrospective nature of this study.

Optimization process

The RNA Toolbox requires optimization regarding background subtraction, image correction, filtering, and image segmentation.

Background optimization

Background subtraction corrects noisy RNA images, but always requires optimization. To analyze the effect of background spatial location, we wrote a MATLAB code that automatically drew six background regions of interest (ROI) in six different spatial regions outside the heart area (Fig. S1). Thirty-two images were analyzed to define the best region from which to draw the background ROI and further optimize the RNA Toolbox. After multiple trials, a background width of four pixels was observed to produce the mean gray values most representative of the background around the heart. Therefore, the background width was standardized to four pixels in all scenarios. We compared the background mean gray values with each other and with those produced from clinical software (eSoft®) using the Pearson correlation. The background in eSoft® was automatically drawn close to the distal boundaries of the LV (location 1 in Fig. S1). A repeated-measures analysis of variance (ANOVA) was performed to compare the mean gray values of the corresponding backgrounds to account for the random effect of the individual.

Filter optimization

This step aimed to evaluate the effects of different filters on the three parameters, LVEF, E, and ApEn, to optimize image processing and filtering using the RNA Toolbox. These parameters were selected to evaluate the performance of the segmentation techniques for each filter scenario. This assesses the filtering within the segmentation context. Four filters (median, mean, Wiener, and median-modified Wiener filter (MMWF)) were compared under two different simulations: 3 × 3 and 5 × 5 kernels. For this purpose, repeated-measures ANOVA and intraclass correlations were used to compare the different filters. Finally, a repeated-measures ANOVA was performed to evaluate the effects of changing filter type and corresponding kernel size on E, ApEn, and LVEF. The eSoft® filter was set to a Butterworth filter with a cut-off of 0.4 and an order of 5.

Optimization of segmentation technique

We developed a novel technique to accurately segment the LV in an RNA scan consisting of 16 sequential frames representing the mean heart volume over time. LVEF was measured using custom MATLAB® code utilizing the novel segmentation algorithms (ACRG). The ACRG used phase and amplitude images because these provide better contrast than spatial domain RNA images. The steps utilized in the ACRG for LV segmentation are outlined below.

An initial seed point was manually selected as the starting point for calculating the initial contour using the RG algorithm. This produces 16 initial contours over 16 frames, which typically extend outside the LV. Using these contours and depending on the grey values within them in each frame, the AC algorithm begins to work on the phase image.

In the phase image, using the AC algorithm, the initial contour evolves over time (in a shrinking pattern) until it finally reaches the LV boundaries in each frame. A set of new contours (i.e., 16 masks) was generated for the next step.

The new contours were applied to the amplitude image to define the maximum boundaries for the RG algorithm. Subsequently, the algorithm began to evolve using the same original seed point from the first step.

Following this, even more precise contours were generated and applied to the actual image to calculate volumes and corresponding activities. Step 2c in Fig. 1 illustrates this occurrence.

LVEFs derived using this novel segmentation algorithm were compared with readings from clinically validated eSoft® software using appropriate methods to address consistency and reproducibility. Agreement between the eSoft® and MATLAB® LVEFs was evaluated using three numerically based processes: intraclass correlation coefficient (ICC), Bland & Altman diagram, and Lin’s concordance correlation coefficient (LCCC).

Predictive models

Continuous variables were expressed as mean ± standard deviation for normally distributed data and as median with interquartile range for non-normally distributed data. Univariate analysis was performed using linear regression to assess the association of outcome (LVEF) with the following predictors: E, ApEn, bounded-ApEn, S, LHR, circularity (diastolic and systolic), elongation (elongation of diastole and systole), short/long-axis FS (diastolic and systolic FS), mean beat duration, rejected/accepted beats, heart rate, or phase angle. Factors associated with LVEF (p < 0.2) in unadjusted univariate linear regression were included in a stepwise multivariate model to identify independent factors associated with LVEF, and the variables were assessed for their usefulness in predicting LVEF. The coefficient of determination (R2) was calculated to determine the proportion of variation in LVEF explained by a set of independent variables Depending on the data, an appropriate post-hoc analysis was conducted to determine the significance of each levels of a variable and the p-values from a test were adjusted by employing Bonferroni correction to account for multiple comparisons. Estimates of beta coefficients and 95% confidence intervals (CIs) were evaluated to determine the extent of association for each independent variable. Linearity and homoscedasticity were examined using scatter and Q-Q of standardized residual plots, as well as Kolmogorov–Smirnov and Shapiro–Wilk tests. The multicollinearity between predictor variables was evaluated using the variance inflation factor (VIF), with values greater than 10 indicating its presence. A multiple linear regression model was used to evaluate the relationship between post-chemotherapy LVEF and all other predictors, derived from the pre-chemotherapy RNA scans. Additionally, the novel RNA Toolbox was used to analyze 103 RNA scans to evaluate the receiver operator characteristic (ROC) curves. See Fig. 1 for further insights into the methods used in this study. All analyses were conducted using SPSS for Windows (version 27; Chicago, IL, USA).

Results

Patient characteristics

The median age of the 103 patients included in this study was 37 (20–81) years. All patients had either stage 2 (n = 79; 39%) or stage 3 (n = 24; 61%) breast cancer with no pre-therapy history of cardiac abnormalities. The corresponding chemotherapy dosages are summarized in Table S2.

Image background optimization

Image backgrounds (Bkg) 1–6 were automatically drawn around the heart along the axis, as shown in Fig. S1. The mean gray values between at least three groups (F (3.5, 415) =10.3, P < 0.001) were statistically significant. Pairwise comparisons revealed that the difference in the mean gray value counts between Bkg1 and Bkg2 (P < 0.05), Bkg1 vs. Bkg6 (P < 0.05), and Bkg2 vs. Bkg3 (P < 0.001) were statistically significant (Table 2). Table S3 shows mean gray values of the corresponding backgrounds with standard deviations. When comparing experimental backgrounds (i.e., Bkgs 1–6) with the eSoft® Bkg, the lowest P-value was found between eSoft® Bkg and Bkg2. No significant differences were observed between eSoft® Bkg and Bkg1, Bkg3, Bkg4, Bkg5, or Bkg6 (P = 0.98). Overall, the results suggest that a change in background spatial locations substantially affects mean gray value. Pearson’s correlation was used to analyze the associations between different backgrounds. The inter-background correlation matrix depicts various correlation values between the ROIs, indicating that changing the background ROI location may affect mean gray values. The strongest correlation was observed between the eSoft® background ROI and Bkg1 (Table S4).

Table 2 Pairwise comparisons showing significant differences between backgrounds (Bkg).

Adjustment for multiple comparisons: Bonferroni. The significance level is 0.05.

Bkg (I)	Bkg (J)	Mean difference
(I-J)	Std. error	Sig.a	95% Confidence interval for difference	
					Lower bound	Upper bound	
Bkg1	Bkg2	2.225*	0.350	<0.001	1.166	3.284	
	Bkg6	1.162*	0.348	0.019	0.109	2.214	
Bkg2	Bkg1	–2.225*	0.350	<0.001	−3.284	−1.166	
	Bkg3	–2.721*	0.452	<0.001	−4.087	−1.354	
	Bkg5	–1.495*	0.359	<0.001	−2.581	−0.410	
Bkg3	Bkg2	2.721*	0.452	<0.001	1.354	4.087	
	Bkg6	1.657*	0.486	0.015	0.190	3.125	
Bkg4	Bkg2	1.542*	0.444	0.012	0.199	2.884	
Bkg5	Bkg2	1.495*	0.359	0.001	0.410	2.581	
Bkg6	Bkg1	–1.162*	0.348	0.019	−2.214	−0.109	
	Bkg3	–1.657*	0.486	0.015	−3.125	−0.190	
Note:

Sig.a, significance; Bkg is the corresponding background; *The mean difference is significant at the 0.05 level.

Filter optimization: a comparison of median, mean, Wiener, and MMWF filters

Our null hypothesis was that the distributions of the median, MMWF, Wiener, mean, median3, MMWF3, Wiener3, and mean3 would be the same. Filter names without numbers (e.g., median filter) represent filters with a 5×5kernel size, whereas filter names with “3” as a suffix (e.g., median3 filter) represent filters with a 3×3 kernel size. Friedman’s test of differences among repeated measures was conducted, and a chi-squared (X2) value of 28 were obtained. A statistically significant difference was observed in the LVEF distributions with different filters (X2(7) = 28; P = 0.001).

Post-hoc analysis using Wilcoxon signed-rank tests was conducted, with the Bonferroni correction applied at all significance levels. The descriptive statistics for all filters and the corresponding kernel sizes are summarized in Table 3. Significant differences were observed between the mean3 and median filters (Z = 2.11, P = 0.002) and between the Wiener3 and median filters (Z = −2.08, P = 0.002). Despite an overall change in LVEF, no significant difference in the LVEF values was observed. Moreover, all significant differences appeared to arise between kernel sizes rather than filter types. Thus, post-hoc tests were performed individually for each kernel size. The results showed no significant differences between the individual 5 × 5 and 3 × 3 kernel size filters (P < 0.05), suggesting no evidence of a difference in the corresponding LVEF distributions of the filters when kernel parameters were fixed. It may suggest an association of LVEF with kernel size but not with filter type. Spearman’s correlation test was performed between 3×3 and 5×5 filter vs. eSoft® LVEFs. The highest correlation between eSoft® and all experimental LVEFs was recorded for the 5 × 5 median filter (r = 0.82).

Table 3 Descriptive statistics of LVEF values under different filter settings.

The filter’s name without a number represents a 5 × 5 kernel size (e.g., Median) and the filter’s name with the number 3 represents a 3 × 3 kernel size (e.g., Median3).

Filter	Sample number	Mean	Std.a Deviation	Minimum	Maximum	Percentiles	
25th	50th	75th	
Median	43.0	62.3	7.6	53.9	86.8	56.2	60.5	64.7	
MMWFb	43.0	62.1	7.5	53.4	85.2	56.2	61.1	64.4	
Wiener	43.0	62.0	7.4	52.7	84.5	56.4	61.1	64.4	
Mean	43.0	62.0	7.4	52.6	85.5	56.3	61.0	64.4	
Median3	43.0	61.3	7.3	50.7	85.0	55.9	59.7	64.2	
MMWF3	43.0	61.3	7.3	50.9	85.0	56.0	60.1	64.1	
Wiener3	43.0	61.1	7.3	51.1	86.0	55.9	60.0	63.7	
Mean3	43.0	61.1	7.1	51.2	84.0	55.9	60.0	64.0	
Note:

Std.a, standard deviation; MMWFb, Median-modified Wiener filter.

A repeated-measurement ANOVA was performed to evaluate the effect of collectively changing the filter type and corresponding kernel size on entropy (i.e., the interaction effect). As repeated-measures ANOVA assumes independence, normality, and sphericity of the data, these assumptions were tested. The Kolmogorov–Smirnov and Shapiro–Wilk normality tests showed the normal distribution of all variables. Mauchly’s test results showed that our E data did not meet the sphericity assumption (P < 0.001) (Table 4). The Greenhouse–Geisser results identified a statistically significant effect of filter (F(1.2, 126) = 288.446, P < 0.001), kernel size (F(1,42) = 81.147, P < 0.001), and filter × kernel interaction (F(1.953,126) = 126.540, P < 0.001) (Table 5). To follow up on these significant main effects and consider the significant interaction (i.e., filter × kernel size), a paired sample t-test was performed for each filter with the corresponding kernel size. The paired sample statistics are presented in Table 6. A greater difference in the mean E values was observed for the median and mean filters (when changing the kernel size) than for the other filters. Thus, changing the kernel size in the median, mean, and Wiener filters significantly affected the E values (P < 0.001) (Table 7). Both kernel sizes depicted alternating patterns in all filters, revealing a significant difference in the mean entropy values (E) between the filters, except the MMWF (Fig. 2). Thus, the MMWF ( 5×5 kernel) was selected for the remainder of this study.

Table 4 Mauchly’s test of sphericity.

Mauchly’s test of sphericity assesses the null hypothesis that the error covariance matrix of the orthonormalized transformed dependent variables is proportional to an identity matrix.

Within-subjects effecta	Mauchly’s W statistic	Approx. Chi-square	df	Sig.	Epsilonb
Greenhouse-geisser	
Filter	0.036	135.785	5	<0.001	0.400	
Kernel	1.000	0.0	0	<0.001	1.000	
Filter * kernel	0.466	31.126	5	<0.001	0.651	
Notes:

a Design: Intercept Within-Subjects Design: filter + kernel + filter × kernel.

b May be used to adjust the degrees of freedom for the averaged tests of significance. Corrected tests are displayed in the within-subject effects table.

Table 5 Test of within-subjects effects (Greenhouse-Geisser) for entropy.

Source	Type III sum of squares	Dfa	Mean square	F	p-value	
Filter	1.053	1.2	0.87	288.4	<0.001	
Error (Filter)	0.153	50	0.003			
Kernel	0.175	1	0.17	81.1	<0.001	
Error (Kernel)	0.091	42	0.002			
Filter * Kernel	0.080	1.9	0.041	126.5	<0.001	
Error (Filter * Kernel)	0.026	82	0.0001			
Note:

Dfa, degree of freedom.

Table 6 Paired-sample statistics of entropy under various filters.

	Mean	N	Std. deviation	Std. error mean	
Pair 1	Median	0.6881	43	0.10265	0.01565	
Median3	0.7589	43	0.07971	0.01216	
Pair 2	Mean	0.6578	43	0.10827	0.01651	
Mean3	0.7329	43	0.08474	0.01292	
Pair 3	Wiener	0.8027	43	0.07579	0.01156	
Wiener3	0.8373	43	0.05807	0.00885	
Pair 4	MMWF	0.8165	43	0.07125	0.01087	
MMWF3	0.8165	43	0.06425	0.00980	

Table 7 Paired samples test for entropy values.

	Paired differences	Dfe	P-value (2-tailed)	
Ma	SDb	SEMc	95% CId of the difference	
Lower	Upper			
Pair 1	Median-Median3	−0.070	0.044	0.0067	−0.08431	–0.05728	42	<0.001	
Pair 2	Mean-Mean3	−0.075	0.044	0.0068	−0.08894	–0.06115	42	<0.001	
Pair 3	Wiener-Wiener3	−0.034	0.0314	0.00481	−0.04436	–0.02495	42	<0.001	
Pair 4	MMWF-MMWF3	0.00001	0.0244	0.0037	−0.00760	0.00762	42	0.998	
Note:

a Mean; b Stadndard deviation; c Standard error of the mean; d Confidence interval; e Degrees of freedom.

Figure 2 Mean entropy values in corresponding kernel sizes.

Both kernel sizes depict alternating patterns with all filters, suggesting a significant difference in mean entropy between filters (except for the median-modified Wiener filter, MMWF).

Optimizing segmentation: Bland-Altman test (eSoft® vs. ACRG)

The graphs in Fig. 3 show the LVEF obtained from 43 patients using two different methods: eSoft® and the RNA Toolbox using ACRG under four different filters. Under all filter settings, a funnel effect was observed, in which differences were reduced with an increase in LVEF values. Additionally, the mean difference deviated significantly from zero, indicating that the RNA Toolbox ACRG provided systematically higher values than eSoft® for LVEFs (see Table 8). The funnel effect does not indicate disagreement, but rather a distribution effect; in addition, this effect may have occurred due to the limited sample size. Thus, LCCC and ICC tests were performed to check for correlations.

Figure 3 Bland-Altman plots between eSoft® and MATLAB® ACRG under filter settings.

These plots show the difference between the eSoft® and MATLAB® ACRG left ventricular ejection fraction (LVEF) readings vs. the mean of both methods (n = 43). The upper and lower dashed lines represent the limits of agreement, and the middle line shows the mean value of the differences.

Table 8 Bland-Altman test values and corresponding Pearson correlations.

	eSoft® vs. RNA toolbox values	
RNA Toolbox:	LVEF (Median)	LVEF (Mean)	LVEF (Wiener)	LVEF (MMWF)	
Arithmetic mean	0.90	1.2	1.23	1.069	
95% CI	[–0.072 to 1.88]	[0.20–2.21]	[0.25–2.20]	[0.068–2.07]	
P (H0: Mean = 0)	0.068	0.019	0.014	0.036	
Lower limit LM	−5.33	−5.2	−4.97	−5.31	
LM 95% CI	[–7.0 to –3.64]	[–6.93 to –3.46]	[–6.65 to –3.29]	[–7.033 to –3.58]	
Upper limit UL	7.14	7.61	7.44	7.44	
UL 95% CI	[5.45–8.83]	[5.88–9.35]	[5.76–9.12]	[5.72–9.17]	
Pearson	r	0.91	0.9	0.9	0.89	
P-value	<0.001	<0.001	<0.001	<0.001	

Optimizing segmentation: LCCC (eSoft® vs. ACRG)

The estimated slope of the regression line (eSoft® vs. ACRG) passes through the center of data at 0.98 (under all filter settings), suggesting high correlation of eSoft® values and RNA Toolbox ACRG outputs. In addition, an average value of R2 = 0.90 suggested that 90% of the variation in the eSoft® variable could be explained by its linear relationship with the RNA Toolbox ACRG. The average value of the concordance correlation coefficient (0.90) was also close to 1 (Table 9), showing good agreement between the two methods. Moreover, the heatmap in Fig. 4 shows hot spots dispersed and located along the line of best fit, indicating that this method may have good agreement with eSoft®.

Table 9 Results of Lin’s concordance correlation between MATLAB ACRG and eSoft®.

Category Algorithm (Filter) vs. eSoft®	Sample size	Concordance correlation*	95% CI	Pearson*	Bias correction factor	
ACRG (Median) vs. eSoft®	43	0.9	[0.84–0.94]	0.9	0.99	
ACRG (Mean) vs. eSoft®	43	0.9	[0.82–0.94]	0.9	0.98	
ACRG (Wiener) vs. eSoft®	43	0.9	[0.82–0.94]	0.92	0.99	
ACRR (MMWF) vs. eSoft®	43	0.9	[0.83–0.94]	0.91	0.99	
Note:

*All corresponding values were statistically significant (p < 0.001)

Figure 4 Regression plot of LVEF derived from MATLAB® ACRG vs. eSoft®.

Optimizing segmentation: ICC (eSoft® vs. ACRG method)

The reliability of a single rating was estimated (number of scans = 43). The estimated mean value of Pearson’s correlation coefficient of all LVEF readings under different filter settings was 0.9 with a 95% CI [0.97–0.98], which was almost identical to the corresponding LCCC results. As the ICC formula considers systematic effects, an ICC value of 0.95, which is very close to Pearson’s correlation coefficient, was obtained. The relatively high ICC may represent a good agreement between the two methods. Figure 5 shows multiple variables graphs of eSoft®-derived LVEFs vs. RNA Toolbox ACRG-derived LVEFs under the four filter settings.

Figure 5 Multiple variables graph.

This graph shows notched box whisker plots of eSoft®-derived LVEFs vs. MATLAB® ACRG-derived LVEFs under four filter settings.

Regression model

The results of the general regression model (n = 103) showed that 41% of the variance in LVEF could be accounted for by five predictors, collectively, (F (5, 97) = 15, P < 0.001). Looking at the unique individual contribution of the predictors, the results showed that LVEF (derived from phase image) (β=0.09,P=0.001), LV ejection rate (β=4,P=0.03), ApEn (β=11.6,P=0.001), ApEn (D&S) (β=39,P=0.002)and LV systole size (β=0.03,P=0.01) were significantly associated with LVEF. These predictors reported acceptable VIF (<1.4), suggesting no serious multicollinearity concerns. The residual plot of this model shows a relatively symmetrical distribution with a tendency to cluster toward the middle of the plot in a random pattern, indicating a good fit for the linear model (Fig. 6). This suggests that cardiotoxicity resulting from anthracycline chemotherapy in patients with breast cancer may affect LV volume, size, and LV dimensions.

Figure 6 Plot of standardized residual by standardized predicted value.

The analysis of results indicates that count-based and size-based predictors made a significant predictive contribution in the model, with LVEF being notably influential in the model. Time-based predictors, such as rejected beats and mean beat duration have less contribution to the model and were characterized as non-image-based parameters. These criteria were determined based on the inclusion of normal/abnormal values and their respective sources.

Risk stratification for chemotherapy-related cardiotoxicity

All predictors were implemented in an ROC curve, which is a test for evaluating the diagnostic performance of predictors and indicates how much a model is capable of distinguishing between classes We re-evaluated the model’s performance in predicting abnormal LVEF (<54%) (adopted from literature (Hesse et al., 2008)) after chemotherapy and found it to be moderately successful. The ROC curve results showed that the model was able to distinguish between the two groups, that is, normal (>54%) and abnormal (<54%) LVEF) at a moderate level (see Fig. 7 for the corresponding area under the curve (AUC) value). Furthermore, the phase LVEF, LA FS, and LV ejection rate were shown to significantly contribute to the prediction (P < 0.05).

Figure 7 Receiver operator characteristic (ROC) curve.

The curve depicts true positive rate (TP) (TP = Sensitivity) plotted as a function of the false positive rate (FP) (FP = 1–Specificity) in the first (left) and second (right) models.

Discussion

Our research findings align with the existing literature (Jones et al., 2020) which posits that certain overlooked image parameters (e.g., dyssynchrony parameters) derived from MUGA scans can be instrumental in detecting subclinical alterations during breast cancer chemotherapy. Administration of cardiotoxic chemotherapeutics (anthracyclines, trastuzumab, etc.) may disturb normal heart function, leading to myocyte toxicity and eventual premature apoptosis. Monitoring LVEF remains the key identifier of cardiotoxicity, and is considered a potent prognostic indicator in patients with known heart failure (Cardinale et al., 2015; Hesse et al., 2008). However, LVEF is not a decisive parameter for cardiac injury during anthracycline chemotherapy, as the relationship between asymptomatic decrease in LVEF (as a percentage) and the onset of heart failure remains unclear (Cardinale et al., 2015). Additionally, clinical protocols that limit RNA scanning early after the completion of chemotherapy may miss late manifestations of cardiac abnormalities due to cardiotoxicity.

We demonstrated the values of the parameters derived from RNA scans. The results of the RNA Toolbox were reproducible and yielded fewer errors and biases than manual processing. To validate the filtering capability, numerous qualitative and quantitative tests were performed to comprehensively analyze the filters. We aimed to systematically choose the most convenient filter for image enhancement, that would not significantly impact parameter calculation or LV segmentation. We used the studies by Lyra et al. (2014) and Salihin Yusoff & Zakaria (2009) on filters in 2D cardiac images as a reference for the filter optimization process. The comparison of experimentally determined LVEF from the RNA Toolbox with that from eSoft® helped in choosing the best filter. RNA images were corrected for background using an automated process (Hu et al., 2018). The mean count within the background ROI was subtracted from all 16 RNA scan frames. As there is no gold standard for validation, the clinical eSoft® package was used to optimize our code. The steps for optimizing image segmentation were set according to previous studies (Garbay, Chassery & Brugal, 1986; Ma et al., 2009; Yang et al., 2012). We propose a novel segmentation method (ACRG algorithm) that uses amplitude and phase images to delineate the LV in RNA scans. Our results are reproducible and repeatable. We found that the RNA Toolbox was superior to eSoft® and other clinical software, because it provides dysynchron numeric parameters and some specific size-based parameters that are not available in other clinical image processing packages. In addition, the RNA Toolbox simplifies data analysis by exporting all RNA scan parameters to an Excel file. Count-based parameters LVEF, phase image, ApEn, ApEn (D&S), LV phase angle, B-ApEn (D&S), B-ApEn, LV systole count, LHR, E, LV diastole count, and synchrony; time-based parameters rejected beats, frame time, and mean beat duration; and LV size-based parameters LV systole size, diastole elongation, and FS were combined to produce a model to predict decreases in LVEF after cardiotoxic chemotherapy treatment in patients with breast cancer. Certain parameters, such as LVEF and ApEn, were associated with a decrease in LVEF after therapy.

The model results highlighted the prevention of cardiotoxicity during chemotherapy. Crucially, cardiotoxicity should be identified before progressing to complicated cardiac events. In a meta-analysis of 22 relevant randomized controlled trials, Li et al. (2020) found that it was imperative to detect cardiotoxicity and reported the significance of cardioprotective agents in improving LV function. The use of medications such as beta-blockers, aldosterone antagonists, angiotensin-converting enzyme inhibitors, and statins prevented serious cardiac events. Moreover, the study concluded that chemotherapeutic drugs should be systematically classified according to the use of certain cardioprotective agents, as current classifications remain relatively arbitrary (Li et al., 2020). Further, we highlighted the importance of integrating cardiotoxicity detection and prediction with further details, in order to assist physicians in personalizing cardioprotective drugs and ultimately prevent further cardiac damage. Our study demonstrates the importance of using parameters to develop a predictive model that aims to reduce LVEF due to therapy. The study results may ultimately help physicians better understand subclinical consequences and enable personalized cardioprotective treatments for patients.

Limitations

This study was principally limited by its retrospective nature, which makes it prone to misclassification bias. Additional selection bias might have been present as our controls were primarily recruited through convenience sampling, and hence may not represent the general population. The limited sample size and limited accessibility for clinical data were also limitations; therefore, prospective studies are required. This study considered only two scans, which may have exaggerated any bias in the data. The average duration of chemotherapy or biological treatment was 12 months, with a substantial variation in the treatment dosage. However, cardiotoxicity may have occurred after 12 months, and significant data may have been missed. Moreover, the effect of comorbid factors on LVEF reduction following chemotherapy has not been evaluated thoroughly owing to difficulties in accessing a full patient history. Although our segmentation method shows good agreement with the clinical eSoft® software, a larger sample number is recommended for further validation.

Conclusions

The RNA Toolbox (available online; Alenezi, 2022) facilitates cardiotoxicity studies using baseline RNA scans. Our findings suggest that certain parameters derived from LV images could predict LVEF decline in patients undergoing chemotherapy, potentially serving as markers for chemotherapy-related cardiotoxicity. Identifying at-risk patients may help tailor treatment plans and optimize outcomes, potentially including the use of cardioprotective drugs. Further prospective studies with larger sample sizes are needed to develop a comprehensive predictive model for LVEF decline.

Supplemental Information

Supplemental Information 1 A code designed to experiment the affect of changing background size and location.

Supplemental Information 2 Bkg with problems.

Supplemental Information 3 A novel code written to extract and calculate Bounded approximate entropy.

Supplemental Information 4 Contrast to noise ratio.

Supplemental Information 5 Compute synchrony entropy.

Supplemental Information 6 Dialogue box.

Supplemental Information 7 ApEn Slow.

Slow implementation of approximate entropy. Inputs: x: (a 1-d vector) input signal m: (positive integer value) Embedding dimension r: (non-negative real value) Tolerance parameter Output: y: ApEn (y is always defined)

Supplemental Information 8 Degree of smoothness.

Supplemental Information 9 Filtering experiment.

Supplemental Information 10 Filtering experiment 2.

Supplemental Information 11 ApEnBoundEnForUnifron noisy signal.

Experimentally Generating uniform signal then applying two types of noise (pulse and multiplicative) After that compute the ApEn and Bounded ApEn for each one

Supplemental Information 12 The main graphical user interface for RNA toolbox project.

This represents the platform of experimenter to evaluate the coded algorithms.

Supplemental Information 13 A set of RNA toolbox.

This offers a comprehensive framework for researchers to analyze parameters such as the left ventricle ejection fraction, dyssynchrony metrics, and numerous additional factors.

Supplemental Information 14 A set of RNA toolbox commands.

This offers a comprehensive framework for researchers to analyze parameters such as the left ventricle ejection fraction, dyssynchrony metrics, and numerous additional factors.

Supplemental Information 15 Patient raw data and statistical analysis.

Supplemental Information 16 The steps of this article.

Supplemental Information 17 Supplementary tables and figure.

The graphical design was provided by Michal Adam Wlodarczyk. Haidar Almohri, a data scientist, provided a limited statistical consultation.

Additional Information and Declarations

Competing Interests

Author Contributions

Human Ethics

Ethics

Data Availability

The authors declare that they have no competing interests.

Ahmad Alenezi conceived and designed the experiments, performed the experiments, analyzed the data, prepared figures and/or tables, authored or reviewed drafts of the article, and approved the final draft.

Fergus McKiddie conceived and designed the experiments, authored or reviewed drafts of the article, and approved the final draft.

Mintu Nath analyzed the data, authored or reviewed drafts of the article, and approved the final draft.

Ali Mayya analyzed the data, authored or reviewed drafts of the article, and approved the final draft.

Andy Welch conceived and designed the experiments, authored or reviewed drafts of the article, and approved the final draft.

The following information was supplied relating to ethical approvals (i.e., approving body and any reference numbers):

Aberdeen Royal Infirmary, NHS Grampian.

The following information was supplied relating to ethical approvals (i.e., approving body and any reference numbers):

NHS Grampian Hospital, Research and Development Unit.

The following information was supplied regarding data availability:

The code and data is available at Zenodo: Alenezi, A. (2024). RNA ToolBox Cardiotoxicity Detection tool for Breast Cancer Chemotherapy by Alenezi. In Cardiotoxicity detection tool for breast cancer chemotherapy: a retrospective study. Zenodo. https://doi.org/10.5281/zenodo.10983610.

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
