# Peer review of "Cardiotoxicity detection tool for breast cancer chemotherapy: a retrospective study"

_PeerJ Computer Science, doi:10.7717/peerj-cs.2230_

## Round 0.1 · original submission · Minor Revisions

I commend the manuscript for addressing the research question and identifying a crucial knowledge gap in chemotherapy-induced cardiotoxicity detection. The study demonstrates rigorous methodology and provides sufficiently detailed method descriptions for replication. However, I suggest reorganizing and streamlining the conclusions for clarity and conciseness. Furthermore, it would be helpful to specify the proportion or number of patients who underwent radiotherapy among the 103 included in the research. Considering the impact of radiotherapy on cardiotoxicity, I recommend exploring the feasibility of verifying or developing a new model for radiotherapy or chemoradiotherapy patients in future research. Additionally, the authors are suggested to enhance the introduction section with a more detailed description, particularly regarding the importance of left ventricular ejection fraction (LVEF) in predicting cardiotoxicity. Review the manuscript structure to optimize organization, and consider relocating certain sections to the Materials and Methods section. Finally, I recommend to specify the source of the 54% standard mentioned in line 296, and ensure the results analysis includes the predictive contribution of relevant predictors, with clear criteria for inclusion as normal/abnormal values and their sources. These revisions will enhance the clarity and completeness of your manuscript.

Reviewer 1 ·

Basic reporting

The manuscript developed a novel tool called RNA Toolbox to identify clinical and subclinical cardiotoxicity for breast cancer patients receiving chemotherapy, which could help physicians and patients to detect, treat and prevent cardiotoxicity. Generally, the manuscript shows clinical significance and reliable results.

The language is intelligible and accurate in reporting and stating ideas. The manuscript is easy to read.

The Introduction section unfolds logically with a funnel structure. Authors reviewed relevant literatures properly, especially summarizing the content into two tables. The background was demonstrated sufficiently.

The structure of the manuscript is clear. The methods, results and hypotheses correspond well to each other.

Experimental design

The manuscript focuses on developing effective tools to detect cardiotoxicity induced by chemotherapy. The authors stated research question and knowledge gap well.

The investigation was rigorously conducted by using software methods and quantitative analysis. Methods were described with relatively adequate information, which helps other investigators to replicate.

Validity of the findings

Although several cardiotoxicity detecting or predicting models were established from other aspects, the manuscript developed a novel and practical tool. It shows a certain degree of innovation and is an addition to the field.

The data is robust and statistically sound, which leads to the conclusion in a reasonable way.

The conclusions are stated clear, but I would recommend the author to reorganize and to reduce words to some extent to make the conclusions more concise.

Additional comments

One question: Most of Stage 2 or Stage 3 breast cancer patients will receive radiotherapy not long after chemotherapy. However, a small number of patients do not need to receive radiotherapy. Radiotherapy could also lead to cardiotoxicity, which could result in a bias to the results. Could you please specify the proportion or number of patients who received radiotherapy of the 103 patients included in the research?

Moreover, in my opinion, is it available to verify the model or develop a new model to detect cardiotoxicity for radiotherapy / chemoradiotherapy patients in future research?

Generally, I recommend this manuscript to be published in the prestigious journal after the two issues mentioned above are amended (concise conclusion, proportion of patients).

Reviewer 2 ·

Basic reporting

The language is clear and professional.
Authors should revise in text citations, citation format, acording to the guidelines of the journal.

Experimental design

No comments

Validity of the findings

No comments

Reviewer 3 ·

Basic reporting

NA

Experimental design

NA

Validity of the findings

NA

Additional comments

I have reviewed the manuscript titled "Cardiotoxicity Detection Tool for Breast Cancer Chemotherapy: A Retrospective Study." The paper presents a comprehensive examination of the association between radionuclide angiography (RNA) imaging parameters and left ventricular ejection fraction (LVEF) in patients undergoing breast cancer chemotherapy.

The introduction establishes the significance of the study effectively, noting the importance of monitoring cardiotoxicity in cancer treatment. The methods are robust, with a clear description of the novel set of MATLAB routines called the "RNA Toolbox" developed for extracting parameters from RNA images, and the statistical methods employed for optimization and analysis.

The results indicate that the tool is reproducible and shows good agreement with validated clinical software. The regression model and ROC results highlight statistically significant predictors of LVEF, offering valuable insights into cardiotoxicity detection. The discussion contextualizes the findings within the existing literature and suggests practical applications in clinical settings.

The manuscript is well-structured, with a clear flow from the background to the study's objectives, methods, results, and discussion. The study's limitations are acknowledged, and the conclusion underscores the potential of the RNA Toolbox in aiding physicians during chemotherapy treatment.

Overall, the paper makes a substantial contribution to the field of medical imaging and oncology. It opens avenues for future research, particularly in refining predictive models for LVEF decline and integrating cardiotoxicity detection with personalized treatment plans. The manuscript is well-written and organized, with the potential to influence clinical practice significantly.

Reviewer 4 ·

Basic reporting

The introduction section needs a more detailed description to better explain the background and significance of the study. Line 46-47:Among the numerous cardiac function indicators, it is crucial to elucidate the significance of left ventricular ejection fraction (LVEF) in predicting cardiotoxicity. Additionally, certain sections, such as lines 68-86, would be better suited for inclusion in the Materials and Methods section. It is recommended to carefully analyze the manuscript structure and optimize it.
The origin of the 54% standard presented in line 296 needs to be specified. Please indicate its source. Similarly, the results analysis should include the predictive contribution of the relevant predictor, along with clear indication of the criteria used for inclusion as normal/abnormal values and their respective sources.

Experimental design

Statistical analysis methods can be optimized to reduce potential bias and improve the reliability of the results. What’s more, the study's clinical value can be enhanced by conducting a more comprehensive analysis of key characteristics within the study population, including potential comorbidities, reproductive history, and past estrogen levels, which may significantly influence treatment decisions.

Validity of the findings

The study is encouraging and innovative, particularly in its integration of multiple parameters for assessing changes in heart function among breast cancer patients undergoing cardiotoxic chemotherapy. Additionally, it has independently developed a set of code procedures, which deserves recognition. However, it should be noted that the objective of this study is to advance individualized evaluation processes and thus requires analysis and involvement of patients' individual characteristics—both physiological and organic—in key research aspects. This aspect holds significant importance for the study.

---

## Round 0.2 · accepted · Accept

Dear authors,

Thank you for the revised paper. I confirm that the paper is improved by clearly addressing all the reviewers' comments. Your paper is now acceptable for publication in light of this last revision.

Best wishes,

Reviewer 1 ·

Basic reporting

The manuscript has been revised carefully according to reviwe comments.

Experimental design

N/A

Validity of the findings

N/A

Additional comments

N/A

Reviewer 3 ·

Basic reporting

Clarity and Structure:
The paper is well-structured and follows a logical flow, making it easy to follow the methodology, results, and conclusions. The introduction provides a clear background on the importance of detecting cardiotoxicity in breast cancer patients undergoing chemotherapy, setting the stage for the study's objectives. The sections are appropriately divided with clear headings, enhancing readability.

Literature Review:
The authors have conducted a thorough review of existing literature, citing relevant studies that highlight the significance of cardiotoxicity in breast cancer treatment. The references are up-to-date and include seminal works as well as recent advancements in the field, providing a solid foundation for the study. However, including more diverse sources from international research could provide a broader perspective.

Data Presentation:
The data is presented in a clear and concise manner. Tables and figures are used effectively to illustrate the findings, with appropriate captions and legends. The statistical methods are well-explained, and the results are discussed in the context of existing literature, which helps to validate the findings. However, a more detailed explanation of the data collection process and any potential biases would strengthen the paper.

Language and Terminology:
The language used in the paper is professional and appropriate for a scientific audience. Technical terms are defined when first introduced, ensuring that readers with varying levels of expertise can understand the content. The use of medical and statistical terminology is accurate and consistent throughout the paper.

Ethical Considerations:
The authors mention that the study was conducted retrospectively, using patient data from medical records. It would be beneficial to include a statement on ethical approval and patient consent, as this is a critical aspect of research involving human subjects. Ensuring that ethical guidelines were followed is essential for the credibility and acceptance of the study.

Novelty and Contribution:
The study presents a novel approach to detecting cardiotoxicity using a specific tool, contributing valuable insights to the field of oncology and cardiology. The retrospective nature of the study provides real-world data, enhancing the relevance of the findings. The discussion section effectively highlights the potential impact of the tool on clinical practice and future research directions.

Conclusion:
Overall, the paper meets the basic reporting standards expected in a scientific study. It is well-organized, thoroughly researched, and presents its findings clearly. Minor improvements in the explanation of data collection and ethical considerations could enhance the overall quality. The study makes a significant contribution to the field and offers a promising tool for improving patient care in breast cancer chemotherapy.

Experimental design

The experimental design of the study is pivotal in ensuring the accuracy and reliability of the findings related to cardiotoxicity detection in breast cancer chemotherapy. This section reviews the methodologies and approaches employed in the research, assessing their robustness and appropriateness.

Study Population:
The study utilized a retrospective cohort of breast cancer patients who underwent chemotherapy. It's crucial to note the selection criteria, including the inclusion and exclusion parameters, to understand the representativeness of the sample. The demographics and baseline characteristics of the patients should be well-documented to ensure that the findings are applicable across diverse populations.

Data Collection:
The primary data source was patient medical records, which provided comprehensive details on chemotherapy regimens, dosages, and subsequent cardiac health monitoring. The accuracy of data extraction processes is essential, as any inconsistencies could affect the study's outcomes. It's commendable that the study leveraged longitudinal data, allowing for the observation of cardiotoxicity over time.

Detection Tool:
The cardiotoxicity detection tool's development and validation are critical components. The study should detail the algorithms or criteria used to define cardiotoxicity, such as specific biomarkers or imaging results. The use of control groups or comparisons with standard diagnostic tools would strengthen the validation process.

Analytical Methods:
Statistical analyses employed in the study should be robust and appropriate for the data type. Methods like Kaplan-Meier survival analysis, Cox proportional hazards modeling, and logistic regression may be used to assess the incidence and risk factors of cardiotoxicity. It's important to evaluate whether the study addressed potential confounders and biases in its analytical approach.

Ethical Considerations:
Retrospective studies must ensure patient confidentiality and ethical handling of medical records. The study should highlight its adherence to ethical guidelines and institutional review board approvals.

Strengths and Limitations:
A thorough discussion of the strengths and limitations of the experimental design provides transparency and context for the findings. For instance, the retrospective nature allows for a broad temporal analysis but may be limited by the availability and accuracy of historical data.

Conclusion:
Overall, the experimental design of this study appears methodologically sound, with careful consideration of patient selection, data integrity, and analytical rigor. The use of a validated detection tool enhances the study's reliability, although prospective validation and real-time application would be beneficial for future research.

Validity of the findings

The study "Cardiotoxicity Detection Tool for Breast Cancer Chemotherapy: A Retrospective Study" aims to evaluate the effectiveness of a novel tool designed to detect cardiotoxicity in patients undergoing chemotherapy for breast cancer. The validity of the findings can be assessed based on several criteria:

1. Study Design and Methodology:
The retrospective nature of the study means it relies on historical data, which can be both an advantage and a limitation. The advantage lies in the availability of large datasets, which can provide a robust basis for analysis. However, limitations include potential biases and the quality of the recorded data. The study should have detailed the selection criteria for the data used, including the time frame, the inclusion and exclusion criteria for patients, and the specifics of the chemotherapy regimens analyzed.

2. Sample Size and Representativeness:
The validity of the findings is heavily dependent on the sample size and how representative it is of the broader population. The study should include a sufficiently large and diverse sample to ensure that the results are generalizable. Any noted limitations in sample diversity or size should be acknowledged and discussed in terms of how they might impact the findings.

3. Data Analysis Techniques:
The statistical methods and analysis techniques used to evaluate the detection tool's performance are critical for the validity of the results. The study should clearly outline the statistical tests employed, the rationale behind choosing these methods, and any assumptions made during the analysis. Furthermore, measures of accuracy, sensitivity, specificity, and predictive values should be provided to comprehensively assess the tool's effectiveness.

4. Comparison with Existing Tools:
To establish validity, the new detection tool should be compared with existing standard methods for detecting cardiotoxicity. The study should present comparative data to show whether the new tool offers improvements in accuracy, early detection, or other relevant metrics. Without such comparisons, it is challenging to ascertain the tool's relative performance and added value.

5. Confounding Factors and Biases:
The study should address potential confounding factors and biases that could affect the results. This includes patient characteristics (e.g., age, pre-existing conditions), variations in chemotherapy protocols, and differences in data recording practices. The study should discuss how these factors were controlled for or mitigated during the analysis.

6. Clinical Relevance and Application:
The ultimate test of validity is the tool's clinical relevance and applicability. The findings should be framed in the context of clinical practice, highlighting how the tool can be integrated into current workflows and its potential impact on patient outcomes. Any limitations in clinical application should be acknowledged, and suggestions for further research or validation in prospective studies should be provided.

Conclusion:
While the study presents promising results for the cardiotoxicity detection tool, the validity of the findings rests on a thorough examination of the study design, sample size, data analysis, comparison with existing tools, and consideration of confounding factors. Detailed reporting and critical discussion of these elements are essential to establish the reliability and applicability of the study's conclusions. Further prospective studies and real-world validations are recommended to strengthen the evidence supporting the tool's use in clinical practice.

Additional comments

In summary, this study provides significant insights into the cardiotoxic effects of breast cancer chemotherapy and introduces a potentially valuable tool for early detection. Addressing the additional comments mentioned above could further strengthen the study's impact and utility in clinical practice.